# Pushing the Boundaries: Forensic DNA Phenotyping Challenged by Single-Cell Sequencing

**DOI:** 10.3390/genes12091362

**Published:** 2021-08-30

**Authors:** Marta Diepenbroek, Birgit Bayer, Katja Anslinger

**Affiliations:** Department of Forensic Genetics, Institute of Legal Medicine, Ludwig Maximilian University of Munich, Nußbaumstraße 26, 80336 Munich, Germany; birgit.bayer@med.uni-muenchen.de (B.B.); katja.anslinger@med.uni-muenchen.de (K.A.)

**Keywords:** forensic DNA phenotyping, FDP, HIrisPlex-S, DEPArray, ancestry prediction, phenotype prediction, massively parallel sequencing, next-generation sequencing, single-cell genomics, single-cell sequencing, mixture deconvolution, low template DNA, ltDNA

## Abstract

Single-cell sequencing is a fast developing and very promising field; however, it is not commonly used in forensics. The main motivation behind introducing this technology into forensics is to improve mixture deconvolution, especially when a trace consists of the same cell type. Successful studies demonstrate the ability to analyze a mixture by separating single cells and obtaining CE-based STR profiles. This indicates a potential use of the method in other forensic investigations, like forensic DNA phenotyping, in which using mixed traces is not fully recommended. For this study, we collected single-source autopsy blood from which the white cells were first stained and later separated with the DEPArray™ N×T System. Groups of 20, 10, and 5 cells, as well as 20 single cells, were collected and submitted for DNA extraction. Libraries were prepared using the Ion AmpliSeq™ PhenoTrivium Panel, which includes both phenotype (HIrisPlex-S: eye, hair, and skin color) and ancestry-associated SNP-markers. Prior to sequencing, half of the single-cell-based libraries were additionally amplified and purified in order to improve the library concentrations. Ancestry and phenotype analysis resulted in nearly full consensus profiles resulting in correct predictions not only for the cells groups but also for the ten re-amplified single-cell libraries. Our results suggest that sequencing of single cells can be a promising tool used to deconvolute mixed traces submitted for forensic DNA phenotyping.

## 1. Introduction

A single human cell contains around six pg of DNA, which is translated to approximately 2 × 3.3 billion base pairs. Sequencing of this amount of genomic data became possible thanks to the rapid development of next-generation sequencing (NGS) [1]. The progress in the field has recently been called, by Nature, one of the technologies to watch in 2021 [2]. The efficacy of single-cell sequencing in medicine cannot be overestimated, and it demonstrates the possibility for implementation in forensics, where it is not commonly used. The main motivation behind introducing this technology into forensics is the potential to improve mixture deconvolution workflows and interpretation methods. If a trace is a mixture of different biological materials (for example, sperm and epithelium), there are methods to separate cell pools to investigate them separately [3,4,5,6,7,8,9,10]. Problems arise when the mixed trace is too complex or if it consists of the same cell type, such as blood-blood mixtures [11]. Implementing single-cell analyses can improve the efficacy of mixture deconvolution and, therefore, has already gained interest in the forensic community. A cell separation solution is the DEPArray™ N×T System from Menarini Silicon Biosystems, which is widely used in clinical research [12,13,14,15] and is suitable for forensic purposes. Current forensic studies focus on single cell separation followed by STR amplification and detection using capillary electrophoresis [11,16,17,18,19]. Mixtures are commonly analyzed with STR typing, but when using different, still new forensic applications, it requires a special approach. Since forensic DNA phenotyping (physical appearance and ancestry predictions) is becoming popular, as observed with legislation expansion in more countries [20], it is assumed that in some cases, a mixture might be the only trace submitted for phenotyping. Phenotyping workflows and prediction methods still need comprehensive research performed before they can be a fully recommended method for mixed trace analysis [20,21,22,23]. Here, we present an alternative approach to dealing with mixed traces submitted for forensic DNA phenotyping that introduces single-cell separation in order to deconvolute the mixture prior to genotyping and phenotyping. 

## 2. Materials and Methods

For the study, a fresh blood sample (female) was collected during an autopsy performed in the Institute of Legal Medicine Munich. The sampling was approved by the Bioethical Commission from the Ludwig Maximilian University of Munich. The blood was used as a reference sample (1 ng DNA input, prepared as described in Section 2.3), for which the white blood cells were separated and used. 

### 2.1. Staining and Counting the Cells

In order to conduct cell separation, the biological material was first stained with the DEPArray™ Forensic SamplePrep Kit (Menarini Silicon Biosystems, Bologna, Italy), which enables the staining of epithelial cells, leucocytes, and sperm cells. Following the manufacture’s protocol, 5 µL of blood was used, from which the white blood cells were marked with PE (Phycoerythrin) conjugated CD45 antibody as well as with DAPI (4′,6-diamidino-2-phenylindole; to stain the nuclei). The maximum cell input for the DEPArray™ N×T System (Menarini Silicon Biosystems) is 6000 cells; therefore, the sample was analyzed with the Countess™ II FL (Thermo Fisher Scientific, Waltham, MA, USA) automated cell counter and diluted to the recommended concentration.

### 2.2. Sorting the Cells

The DEPArray™ N×T System’s (Menarini Silicon Biosystems) forensic protocol was used to separate the white blood cells. Only highly evaluated (of typical size, morphology, and staining) cells were chosen for the experiment. The selected cells were moved automatically by the DEPArray™ platform and placed in dedicated tubes as follows: 5 groups of 20 cells, 10 groups of 10 cells, 10 groups of 5 cells, and 20 single cells (groups and single cells were collected in two separate experiments).

#### Preventing Contamination

Due to working with a special type of material (single cells), ensuring that there is no contamination is extremely important for the interpretation of the data. Since the DEPArray™ N×T System is a closed system, contamination within is not possible. In order to exclude its potential occurrence in the next stages of laboratory work, all steps were carried out with a negative control included.

### 2.3. Library Preparation and Sequencing

For all samples used in the study, genomic DNA (gDNA) was extracted with the DEPArray™ LysePrep Kit (Menarini Silicon Biosystems) directly from the cell separation tubes to make sure the input corresponds with the number of collected cells. All sample extracts were subjected to manual library preparation using the Precision ID Library Kit and IonCode™ barcode adapters using the validated Ion AmpliSeq™ PhenoTrivium Panel [24]. The amplification of the targets was carried out in the same tubes as the extraction to avoid changing the DNA input. The target amplification cycle number and annealing/extension time were 23 cycles and 4 min, respectively. Libraries prepared from cell groups and half of the collected single cells were quantified using the Ion Library TaqMan Quantitation Kit (Thermo Fisher Scientific), diluted (except for libraries of <30 pM concentration), and pooled equimolarly to 30 pM for template preparation on the Ion Chef using the Ion S5™ Precision ID Chef & Sequencing Kit (Thermo Fisher Scientific). The remaining half of the single-cell-based libraries were quantified and rescued by library amplification (see Section 2.3.1). A range of 12–24 libraries were pooled per 530 chip and sequenced on the Ion S5 [25]. 

#### 2.3.1. Library Amplification

Half of the undiluted single-cell-based libraries were submitted for library amplification. A 25 µl sample of the previously eluted samples was combined with 72 μL of Platinum™ PCR SuperMix HiFi and 3 μL of Library Amplification Primer Mix from the Precision ID Library Kit (Thermo Fisher Scientific). The PCR products were purified in a two-round clean-up with the Agencourt™ AMPure™ XP Reagent according to the manufacturer’s manual. The amplified libraries were quantified, diluted, and pooled for automated template preparation.

### 2.4. Data Analysis

Primary sequence analysis was performed on Torrent Suite™ Software (TSS) 5.10.1 (Thermo Fisher Scientific) with Torrent Mapping Alignment Program (TMAP) alignment of sample reads against the hg19 genome assembly. SNP genotyping and tertiary analysis, in the form of ancestry prediction, were performed using the HIDGenotyper–2.2 plugin and Converge v2.2 (Thermo Fisher Scientific). Low-quality results were double-checked by running VariantCaller v5.10.0.18 on TSS and reviewing the raw data in IGV 2.7 (Integrative Genomics Viewer) [26]. Tertiary analysis was separated into two parts: phenotype prediction (which cannot be performed within Converge v2.2) and ancestry prediction by the bootstrapping admixture analysis with Converge. For the cell groups, the coverage thresholds were adjusted as follows: for the SNPs corresponding with the HIrisPlex-S panel, the analytical coverage thresholds were set based on the HIrisPlex-S panel validation for MPS platforms [22]; for the remaining autosomal ancestry markers, the minimum coverage to call an SNP was set to 100 reads. For the single cells, the coverage threshold was lowered to 50 reads for all markers. The heterozygote balance threshold was set to 65%/35% for heterozygotes and 90%/10% for homozygotes. 

#### 2.4.1. Phenotype and Ancestry Prediction

For phenotype (HIrisPlex-S: eye, hair, and skin color) prediction, SNP genotypes were exported from Converge and used to manually generate single profiles, as described in Section 2.4.2, which were later converted into the input file format required by the HIrisPlex-S Webtool (https://hirisplex.erasmusmc.nl/ accessed date 29 April 2021). The HIrisPlex-S SNP set contains an indel SNP (rs796296176) in the form of an A insertion that was manually reviewed and called using IGV 2.7. Predictions were interpreted according to the PhenoTrivium validation paper [24]. For ancestry prediction, called genotypes were merged into a consensus single SNP profile using Converge, and the prediction was performed with the bootstrapping admixture analysis [27] feature of Converge using the 75% resampling size, 1000 replications, and the Precision ID Ancestry Panel Ancestry Frequency File v1.1. The PhenoTrivium Panel contains the 145 Precision ID Ancestry SNPs used for bootstrapping admixture analysis. In the bootstrapping admixture analysis feature of Converge, admixture predictions are made based on a maximum likelihood approach used to predict the most likely admixture proportions across seven root populations (herein referred to as the core admixture algorithm): Africa (AFR), East Asia (EA), South Asia (SA), Southwest Asia (SWA), Europe (EU), America (AME), and Oceania (OCE). The predictions are bootstrapped across a random subset of sequenced SNPs, specified by the user in %, with each bootstrapping replication ran through the core admixture algorithm N times using a different subset of SNPs for each replication to capture uncertainty in the predictions. The results are displayed as an average of the bootstrapping replications for each population group and a 95% confidence interval reflecting the probable range of variability of the estimated ethnicity percentages. 

#### 2.4.2. Interpretation Models

The consensus genotypes from cell groups and single cells were used to generate a single SNP profile for tertiary analysis. Two different interpretation models were used to build the profiles: “basic”, which compromised genotypes detected at least twice, and “conservative”, for which genotypes were included only if detected at least four times. The cell groups were interpreted using only the “basic” approach (following the interpretation pipeline from PhenoTrivium validation), and the profiles from the single cells were generated using both models. Additionally, the tertiary analysis was performed based on single profiles obtained from single cells with the highest coverage. 

## 3. Results

### 3.1. Cell Groups

The cell groups were collected to evaluate the performance of the workflow combining single-cell separation using the DEPArray™ N×T System with Next Generation Sequencing using the Ion S5. The results obtained in this study were compared to the data from a previously performed validation study [24] of the PhenoTrivium assay. The collected cell groups approximately corresponded to 100, 50, and 25 pg DNA, which were similar to the DNA input used in the sensitivity study (125, 62, and 31 pg). 

#### 3.1.1. Coverage, Allele Frequency, and Genotype Calling 

The study consisted of 5 groups of 20 cells, 9 groups of 10 cells (one sample was excluded due to a library prep error), and 10 groups of 5 cells, for a pool of 24 libraries total (one group of cells corresponds to one library), which was sequenced on a 530 Chip. Marker coverage across the 200 autosomal markers included in the panel ranged between 309,357 and 736,887 total reads for 20 cells, 216,401–549,980 reads for 10 cells, and 106,865–367,782 reads for 5 cells. The obtained values were comparable with the ones observed in the sensitivity study, where the total number of reads across the triplicates was 340,461–698,760, 374,692–637,041, and 94,120–395,664 for 125, 62, and 31 pg respectively [24]. The coverage for each marker is presented in Appendix A.

Allele frequencies were calculated for 43 out of 45 heterozygote loci. Markers rs1470608 and rs10756819 (from HIrisPlex-S) were not called among all the samples due to low coverage. Frequencies were calculated by dividing the number of reads obtained for the reference allele by the total number of reads per locus obtained for both reference and alternative allele. For 20 cells, the frequency varied between 45% and 57% (average 50%), for 10 cells between 47% and 55% (avg. 51%), and for 5 cells between 43% and 58% (avg. 50%). The allele frequency for loci classified as homozygote was between 92–100% for 20 cells (avg. 99.8%), between 90% and 100% for 10 cells (avg. 99.6%) and between 87% and 100% for 5 cells (avg. 99.5%).

Previously described coverage and heterozygote balance thresholds were used to call genotypes in Converge using the HID Genotyper–2.2 plugin. For 20 cells, between 94 and 99.5% of genotypes across all 200 autosomal loci met all the thresholds. For ten cells, the values were 84–98.5%, and for five cells, 64.5–98.5%. The percentage of called genotypes in the sensitivity study was 95–98.5% for 125 pg, 88.5–90% for 62 pg, and 72–81.5% for 31 pg. All the called genotypes were in concordance with the reference sample, no incorrect genotypes were observed. In comparison, the data from the previous study showed that allelic dropouts and drop-ins occurred among the replicates with a 31 pg DNA input, which caused incorrect genotype calling. The detailed genotyping results for all the samples are presented in the Appendix A.

#### 3.1.2. Phenotype and Ancestry Prediction

Phenotype and ancestry predictions were performed for the reference sample and for ‘basic’ consensus profiles generated for cell groups. From 41 markers associated with eye, hair, and skin color prediction, for all cell groups, 39 SNPs were recovered, and two loci were not called due to low coverage, namely rs1470608 and rs10756819 (similar as in the sensitivity study). Observed area under curve (AUC) accuracy loss (0.001 for very pale, 0.004 for pale, and 0.001 for intermediate skin) did not affect the final phenotype prediction, which was predicted to be the same as the reference, with the following *p*-values: brown eyes (0.998), black hair (0.661 for black and 0.995 for dark), and dark skin (0.696). From 145 markers used by Converge for ancestry analysis, the maximum number of SNPs was obtained from 5 cells while the profiles for 20 and 10 cells included one locus not called due to low coverage (rs2196051). The bootstrapping admixture analysis for all cell groups revealed an admixture of Southwest Asia and Europe (ratio ca. 70/30%), and the reported individual’s place of birth was Iraq (no further details available). The detailed prediction results for all the samples are presented in Table 1.

### 3.2. Single Cells

#### 3.2.1. Coverage, allele frequency, base misincorporation rates, and genotype calling 

For the unamplified libraries (NA), the number of mapped reads varied between 11,979 and 58,536, of which approximately 60% were on target (Figure 1a). Total coverage across all the markers analyzed in Converge ranged between 1891–26,400 (avg. 8889) reads, and the mean marker coverage ranged between 9 and 132 reads (avg. 44). For the amplified libraries (AMP), the number of mapped reads increased to a range of 44,445 to 186,402 reads, of which ca. 60% were on target (Figure 1b). This resulted in total coverage between 13,735 and 113,398 reads (avg. 49,642) and mean marker coverage between 69 and 567 reads (avg. 248). The coverage for each marker is presented in Appendix A. To visualize the coverage difference between the unamplified and amplified libraries, the percentage of amplicons with a very low and very high number of reads were compared (Figure 2). 

Due to a significant number of missing loci observed among the NA libraries, the allele frequency was estimated for the heterozygote loci (except for rs1470608 and rs10756819, which were not called due to low coverage) only among the AMP samples. The mean value per locus varied between 43% and 63%, and the detailed allele distribution across the loci is presented in Figure 3. For most of the markers, the mean heterozygote balance stayed between the accepted threshold of 35% and 65%, and only three markers (rs8035124, rs917115, rs4821004) showed imbalance. The average allele frequency for the remaining expected homozygotic loci was 100%, but for each sample, loci with frequencies below that were observed (Figure 4). The few loci per sample that did not meet the thresholds were marked as imbalanced, which did not result in an incorrect genotype being called. 

Due to the special materials used, which were single cells, sequencing errors could be expected, like non-specific base misincorporation. When reviewing the genotypes in Converge, unexpected nucleotides were detected for two loci, namely rs7722456 (reference: C, variant: T, observed: A with maximum 20% frequency, resulting in incorrect genotyping) and rs8113143 (reference: C, variant: A, observed: G with maximum 3% frequency, discarded). The incorrect nucleotides were not observed after running the variantCaller plugin on Torrent Suite™ Software (TSS) 5.10.1 and when reviewing the raw data with IGV 2.7 (Integrative Genomics Viewer). Incorrect genotypes called in rs7722456 were discussed before [28] and can be explained by the homopolymer region flanking the SNP (as well as for the other SNP discussed, rs8113143). The genotypes for rs7722456 in Converge were called manually.

Previously described heterozygote balance and lowered coverage thresholds (50×) were used to call genotypes in Converge using the HIDGenotyper-2.2 plugin. For NA libraries, only between 2,5% and 48,5% (avg. 26%) genotypes across all 200 autosomal loci met all the thresholds, and for the AMP samples, the range was between 49,5% and 96% (avg. 78%) (Figure 5). Among the dropouts, most were loci with low coverage (78-185 loci for NA and 3-94 loci for AMP), and the remaining were imbalanced (max. 25 per sample for NA and 13 per sample for AMP). Across all samples, no genotype was called for one marker (rs10512572) despite passing the coverage threshold. After running the variantCaller plugin, the genotyping for this locus also failed. Upon reviewing the sequencing data in IGV 2.7, it was observed that leftover primer reads were misaligned, resulting in no call. Genotypes for this locus were called manually. Single incorrect genotypes were observed for both groups, all due to allelic dropouts. The detailed genotyping results for all the samples are presented in the Appendix A. 

#### 3.2.2. Data Interpretation–“Basic” and “Conservative” Models

Due to a significant number of missing loci observed among NA libraries (the Fisher exact test statistic value is < 0.00001), only the profiles from the AMP libraries were submitted for further interpretation. For the “basic” and “conservative” models, consensus profiles from a maximum of 41 phenotypes and 145 ancestry SNPs were generated as described in Section 2.4 and Section 2.4 All single cells had the same genotype called for 13 out of 41 phenotype-associated markers (concordant with the reference), while for ancestry, there were 50 out of 145 markers (Figure 6). Only three incorrect genotypes were observed across all the SNPs, all once per marker, which did not affect the prediction. 

The final “basic” profile consisted of 39 phenotype SNPs (rs1470608 and rs10756819 not called due to low coverage) and 145 ancestry SNPs, which resulted in the same *p*-values for the phenotype prediction and the same admixture (calculated by Converge) as the cell groups (Table 1). The “conservative” model required a genotype to be called at least four times across all the single cells, which left 38 phenotypes and 139 ancestry markers used in the final profile. The one additional missing loci among the phenotype markers, namely rs1545397 (low coverage), caused an insignificant change in the *p*-values obtained for the skin color but overall did not affect the final phenotype prediction (Table 1). The ancestry analysis performed by Converge showed an admixture of Southwest Asia and Europe with a proportion of ca. 60/40%.

#### 3.2.3. Data Interpretation-Single-Cell Based Predictions

From the ten sequenced single cells, six were selected for tertiary analysis due to their high genotype calling rate of ca. 90% (Figure 7). All profiles used for the phenotype prediction were partial with 33–38 SNPs; therefore, the obtained predictions were reported together with calculated AUC loss as recommended by prediction models [29] (Table 2). For eye color, all cells resulted in the correct brown color prediction, where one sample showed a slightly different *p*-value than expected. The correctness of eye color predictions is strongly dependent on the rs12913832 marker [30,31], and this locus was correctly genotyped among all the single cells. For hair color, the obtained *p*-values were identical or almost identical as for the reference for all except one cell. For the sample named ID4, the highest *p*-value was brown with 0.521, while black was 0.465. The profile missed the rs16891982 marker (imbalanced), for which the correct CC genotype is strongly associated with black hair color [32,33]. Despite the difference, the final prediction was black hair color for all samples. The biggest variance was observed for skin color due to several markers relevant for prediction missing. Furthermore, skin color is the most complex phenotypic trait to predict, and incomplete profiles make it even more difficult [34,35]. For only three cells, the final prediction would be the same as the reference, namely dark skin with 70% probability. For samples ID4, ID7, and ID10, the *p*-value for dark skin was only around 0.5, and for the last two samples (ID7 and ID10), the second-highest was dark to black, which would incorrectly suggest the skin color to be darker. 

The profiles used for ancestry prediction were also incomplete, which compromised 120 and 140 SNPs. For all the samples, the analysis performed by Converge revealed an admixture of Southwest Asia and Europe, and for all except one sample, the proportion was close to what was detected for the reference. The result obtained for sample ID4 suggested more European than Southwest Asian ancestry and was caused by missing genotypes for rs16891982 and rs2196051, which are strongly associated with South Asian, not European, populations.

## 4. Discussion

Mixture deconvolution is one of the most engaging topics in forensics. Over previous years different methods have been developed to assist the process, both prior to [3,4,5,6,7,8,9,10] and after, DNA typing of a mixed trace [36,37,38]. The DEPArray System, which uses a dielectrophoresis grid to isolate single cells, is a promising cell separation technology for forensic analysis. It was demonstrated that the DNA from the collected cells can successfully be processed with STR amplification kits and detected using capillary electrophoresis [11,17,18,39]. Single cells are considered ltDNA (low-template DNA), and handling such material is not uncommon in forensics, where specialists are often faced with low-quantity and low-quality DNA. Since such samples are treated with special lab protocols (e.g., increased number of PCR cycles), it makes them prone to stochastic effects (drop-ins and dropouts) and increased stutter ratios, the interpretation of ltDNA profiles is a common topic of discussion [40,41,42,43,44,45]. An alternative to traditional fragment analysis is to sequence the bi-allelic SNPs, which are less prone to artifacts and are characterized by smaller amplicons. The analysis of SNP markers is becoming more and more popular in forensics, especially thanks to the rapid development of massively parallel sequencing (MPS), which is not only more sensitive, but also features the use of high multiplex panels consisting of hundreds of markers. Successful single-cell-based STR experiments suggest additional applications in other forensic investigations. A combination of single-cell analysis and SNP sequencing could be especially useful for forensic DNA phenotyping and serve as an alternative solution for the analysis of mixed samples, for which the current interpretation of the phenotype and ancestry predictions can be complicated or impossible.

Panels containing SNPs associated with phenotype [22,46,47,48] and ancestry [21,49,50,51] have been introduced, and recently assays combining those have been published [23,24,52]. Forensic DNA phenotyping is now legislated in different European countries [20], and it is expected that police investigators will seek expert opinions on samples of interest. The desirable interpretation of phenotypic features should be based on traces derived from single individuals because mixed samples might not provide reliable results. The collection of single-source traces may not always be possible; therefore, published studies on phenotypic/ancestry SNPs should also discuss the analysis of mixtures. The first and basic indication of a mixed SNP profile is an overall increase in the number of heterozygotic loci [21,53]. This statement cannot easily be applied for individuals originating from admixed populations (or having biparental co-ancestry) where the number of heterozygotes is naturally higher. However, as presented in the study by Eduardoff et al. [21], the allele read frequencies (ARF) differ between unmixed and mixed samples. Mixtures exhibit a higher number of heterozygotes not meeting the balance threshold. Although the authors were able to successfully detect the major and minor contributors for all tested mixture ratios (from 1:1 to 1:9), they recommend careful data analysis, especially for extreme mixtures. The authors of the HirisPlex-S MPS validation paper [22] designed a calculator to aid an interpretation of 2-person mixtures, which is based on known major/minor contributor ratios obtained from STR typing done prior to SNP analysis. The ratios are used to separate read counts obtained during sequencing. The approach was tested on mixtures with different ratios (from 1:1 to 1:9) and using contributors with distinguishable genotypes and phenotypes. The exercise shows that the variants for 28 of the 41 SNPs included in the HirisPlex-S were successfully separated into individual profiles. The validation study of the comprehensive assay combining markers for appearance and ancestry predictions from the VISAGE consortium [23] also mentions mixture deconvolution, similarly of two contributors with different phenotypes and biogeographical ancestries, where the analysis was based on allele frequencies. A publication from Ralf and Kayser [54] presents the first actual crime scene trace of a mixed source submitted for phenotype prediction. The 2-person mixture sample of interest had almost 500 pg DNA. The phenotyping was done using the MPS-based HirisPlex-S Panel, and the sequencing included both the trace and reference material obtained from the victim. The authors were able to extract the probable suspect’s genotype for all the SNP markers (one with an alternative genotype considered) and obtained distinguishable phenotypes with high probability values for all predicted traits. They also performed mixture deconvolution by using the aforementioned tool by Breslin et al. [22] and revealed concordance for 36/41 tested SNPs. The remaining genotypes were interpreted differently using both approaches. Overall, discussed studies show that the interpretation of the mixed samples must be made with caution, and the approach should be further evaluated, especially for low input samples or mixtures of individuals with indistinguishable phenotype/ancestry.

The advantage of implementing mixture deconvolution methods prior to the sequencing of the phenotypic/ancestry SNPs is being able to perform the predictions based on single-source profiles. In the case of single cells, the reliability of those predictions will depend on the quality of the obtained sequencing results. As already discussed before, the ltDNA requires careful and cautious analysis, and even though SNPs are less prone to stochastic effects, the following issues are expected when sequencing low template material: allelic drop-ins and dropouts, locus dropouts, imbalanced genotypes, and increased base misincorporation rates. The sensitivity studies of the large SNP assay suggest that the threshold for two of the most popular MPS platforms used in forensics, namely MiSeq FGx System and Ion S5, lies around 100 pg DNA [22,23,24,52]. In all studies, inputs as low as 5–7 pg were assessed, which roughly corresponds to one human cell. As expected, with such a low input, the obtained results consisted largely of no calls, drops in/out, and incorrect calls. Different forensically relevant SNP panels were tested for their efficacy in typing ltDNA, both using the golden forensic detection standard, namely capillary electrophoresis [55] and MPS [56,57]. Assuming that the DNA input is low, additional changes might be introduced to enhance the standard protocols to boost the coverage. The most common adjustment is to significantly increase the number of PCR cycles used to amplify the targets. The expected positive effect is a higher coverage, but some issues can also occur, like allelic drops in/out causing imbalances or incorrect calls [55,56]. When working with MPS and the AmpliSeq pipeline library, amplification might be applied in order to improve the read depth. The libraries which yielded low quants might be amplified with a high-fidelity PCR supermix. This step in the library preparation was previously implemented when using the AmpliSeq workflow in other molecular fields [58,59,60,61], but it does not seem to be a common practice in forensics. A study by Meiklejohn and Robertson mentions the amplification of libraries, but only in order to quantify them after using Qubit and Bioanalyzer [62]. The potential effect of the library amplification on the low-template samples is presented by Turchi et al. [63]. The authors performed a comprehensive validation of the Precision ID Identity Panel by Thermo Fisher Scientific by sequencing challenging forensic samples on the Ion PGM. The libraries from their study with less than 30 pM were amplified and again purified, similarly to our single cell-based libraries discussed in this paper. It was observed that the library amplification step helped to obtain a higher level of repeatability and improved the values of sequencing parameters. Our study shows that introducing this step resulted in over five times higher total coverage of the single cell-based libraries. The number of no-calls per sample decreased from an average of 70% for the unamplified libraries to 18% for the amplified ones. The highest number of imbalanced genotypes per sample was also half as low, and a maximum of two incorrect genotypes per sample were observed after the libraries were amplified. 

The goal of sequencing a trace submitted for forensic DNA phenotyping is to obtain reliable genotypes which will be used for predictions. Incorrect phenotype and ancestry predictions might be caused by incorrect genotypes [64] and incomplete profiles [29,65]. The authors of the HirisPlex-S Panel explain that different genotypes missed from the profile will have different influences on the prediction model and that the incomplete data has to be reported as a probability of each trait predicted together with AUC (area under curve) accuracy loss [22,29,35]. We tested an approach of using incomplete single profiles generated by single cells to observe how they will affect the predictions. The calculated AUC loss would not be considered significant, but the missing genotypes resulted in *p*-values notably different than for the reference samples, causing even incorrect predictions for the skin color. In their paper, Cheung et al. tested how different percentages of genotypes missing from a profile will affect the ancestry prediction when analyzed with different classifiers [65]. Their study shows that an increasing number of missing markers can affect the predictions, especially in the case of admixed samples. Our method to perform the ancestry analysis was not mentioned in the paper, but our results show similar observations. We used bootstrapping admixture analysis based on a maximum likelihood approach to predict the most likely admixture proportions across seven root populations. The SNP profiles were bootstrapped using a varying number of replications, where each replication selected a random subset of SNPs to capture uncertainty in the predictions [27]. For the analysis of the single profiles obtained from the single cells, the number of SNPs available for bootstrapping differed between 123 and 139 out of 145. However, the number of markers missing had less impact than which markers were missing. For samples ID4 and ID8, almost the same number of SNPs were used for the analysis (123 and 125 respectively), but the prediction outcome was an admixture of Southwest Asia and Europe with a difference of 30%. For both phenotype and ancestry, the most accurate predictions in comparison to the reference sample were obtained for the “basic” consensus profile built from single cell-based libraries with the highest genotyping rate. 

## 5. Conclusions

This study is a proof of concept demonstrating that single-cell sequencing can obtain the correct phenotype (HIrisPlex-S) and ancestry prediction, and therefore, can be considered a viable mixture deconvolution method for challenging samples submitted for forensic DNA phenotyping. The presented workflow combined single-cell separation with the DEPArray™ NxT System and sequencing of phenotype and ancestry-associated SNPs with the Ion S5 platform. The DNA typing was done using a previously validated custom Ion AmpliSeq™ PhenoTrivium Panel. The study was based on testing groups of 20, 10, and 5 cells, for which the results were comparable to DNA sensitivity tests performed in a previous validation study, showing the potential of the combined workflow. The number of tested single cells was 20, where half of the libraries were ’rescued’ with library amplification prior to sequencing. Introducing this step helped to significantly increase the total coverage obtained for the 10 amplified libraries, which resulted in recovering more genotypes. More than half of the rescued single cells had a genotyping rate close to, or greater than, 90%. Additionally, different interpretation approaches were evaluated. The predictions for single cells were based on a “basic” and a “conservative” consensus profile (genotype called two or four times, respectively) and on single profiles themselves. The most reliable predictions were obtained when using the “basic” consensus profile, for which almost no dropouts were observed. The results suggest that collecting single cells from a mixed sample prior to forensic DNA phenotyping can be used as an alternative way of performing phenotype and ancestry predictions of the mixture’s contributors. This approach will be further evaluated by sequencing mock mixtures with different ratios using contributors not only with distinguishable phenotype/ancestry but also similar ones. Additional changes to the workflow will also be evaluated, like increasing the number of PCR cycles prior to amplification, adjusting library inputs, and improving the genotyping. 

## Figures and Tables

**Figure 1 genes-12-01362-f001:**
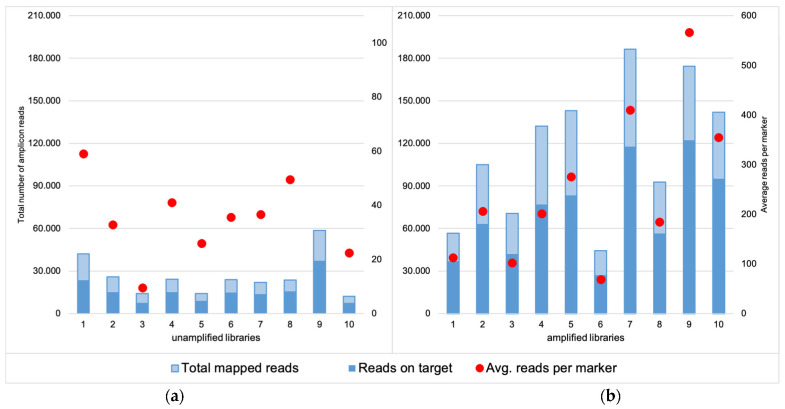
Comparison of the total mapped reads obtained for the unamplified (**a**) and amplified libraries (**b**), together with the number of the reads on target for both groups. The latter was comparable and reached around 60% for both unamplified and amplified libraries but the total number of mapped reads for the amplified libraries was much higher and therefore resulted in higher total coverage. Red dots represent the average coverage per marker for each sample (coverage threshold was set as 50 reads).

**Figure 2 genes-12-01362-f002:**
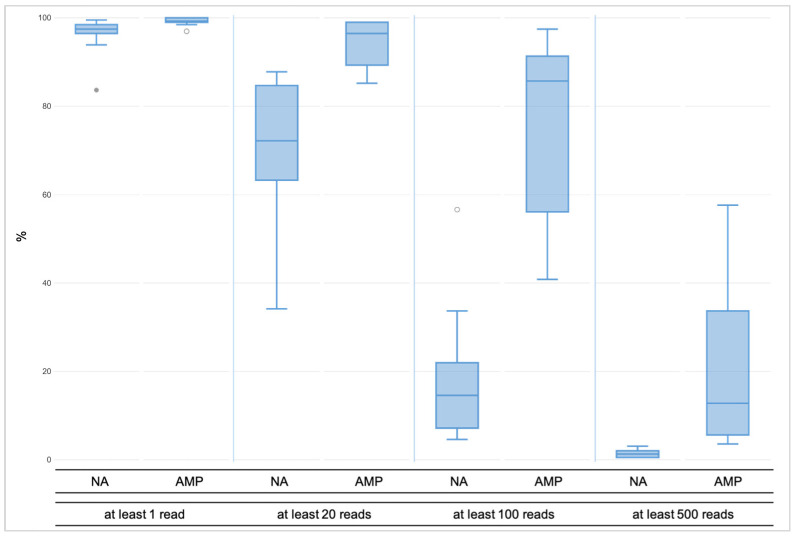
The minimal coverage of at least one read was observed for over 99% of all amplicons across the amplified libraries (AMP), whereas for unamplified libraries (NA), the value was 96%. At least 20 reads were reached by little less, namely 94% of the AMP libraries, while across the NA libraries, it was noticeably less, namely 70% of the amplicons. Over 75% of the amplicons across the AMP libraries had coverage of at least 100 reads, but less than 20% of the NA libraries reached the coverage. The 500X coverage was observed for almost 20% of the amplified libraries and 1% of the NA libraries.

**Figure 3 genes-12-01362-f003:**
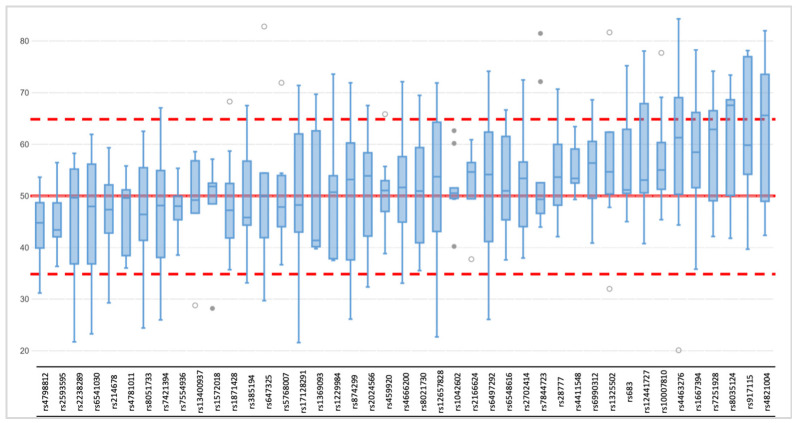
Allele frequency distribution across 43 heterozygote loci (rs1470608 and rs10756819 not called due to low coverage) among ten single-cell based libraries (AMP); red lines correspond to expected (50%), upper (65%), and lower (35%) thresholds.

**Figure 4 genes-12-01362-f004:**
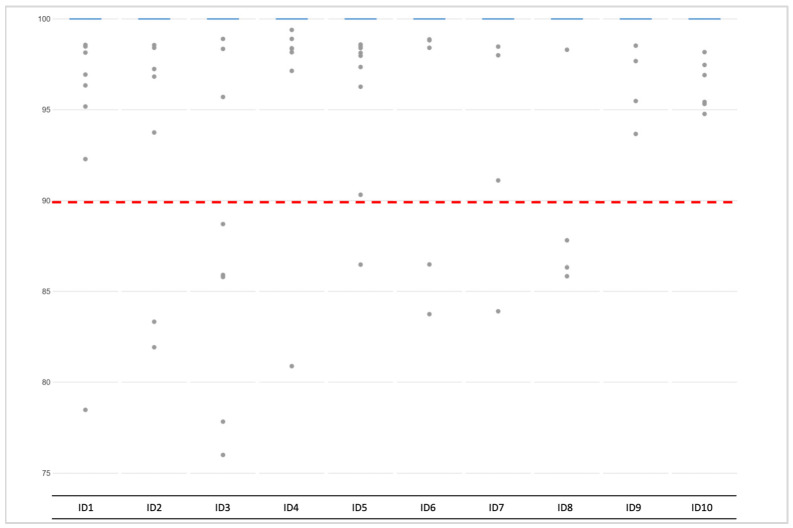
Allele frequency distribution across 155 homozygote loci among ten single-cell-based libraries (AMP). Grey dots represent the homozygotic loci with less than the expected 100% frequency, which is marked with blue lines. Red line corresponds to lowest (90%) accepted threshold. The minimum number of loci per sample below threshold was one; the maximum was five.

**Figure 5 genes-12-01362-f005:**
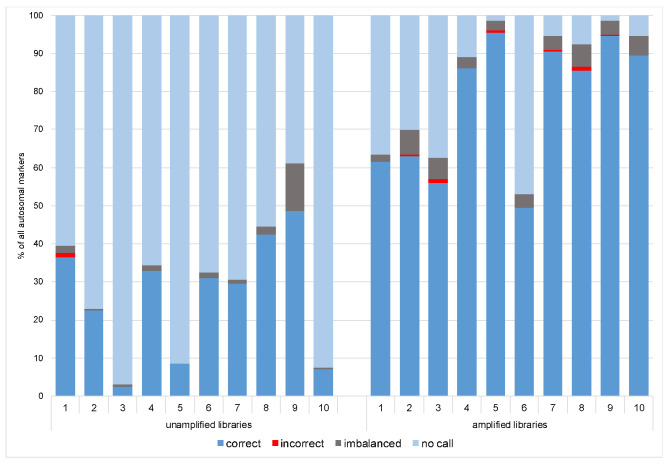
Genotyping rate per each sample across all 200 markers for unamplified and amplified single-cell-based libraries.

**Figure 6 genes-12-01362-f006:**
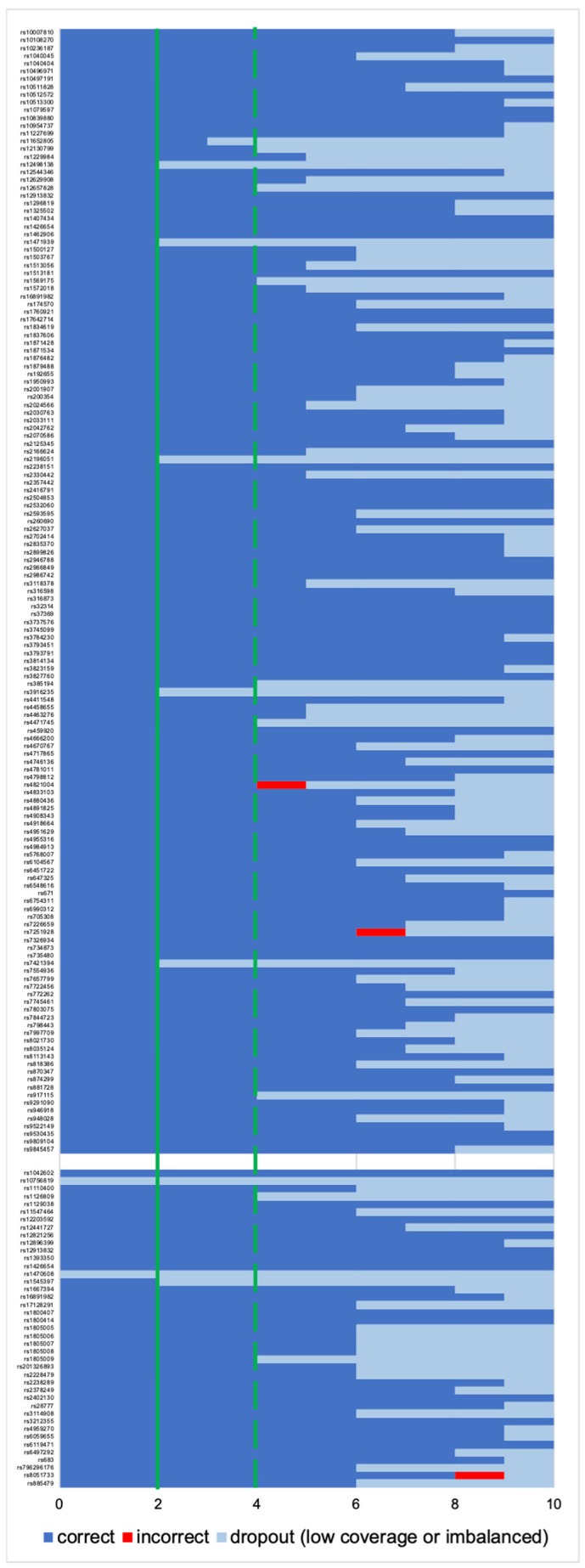
Genotyping rate per each locus used for ancestry prediction (max. 145 SNPs, upper part) and phenotype prediction (max. 41 SNPs, lower part). The x-axis shows the number of single cells with correct, incorrect or no genotype. Solid line: “basic” profile; dotted line: “conservative” profile.

**Figure 7 genes-12-01362-f007:**
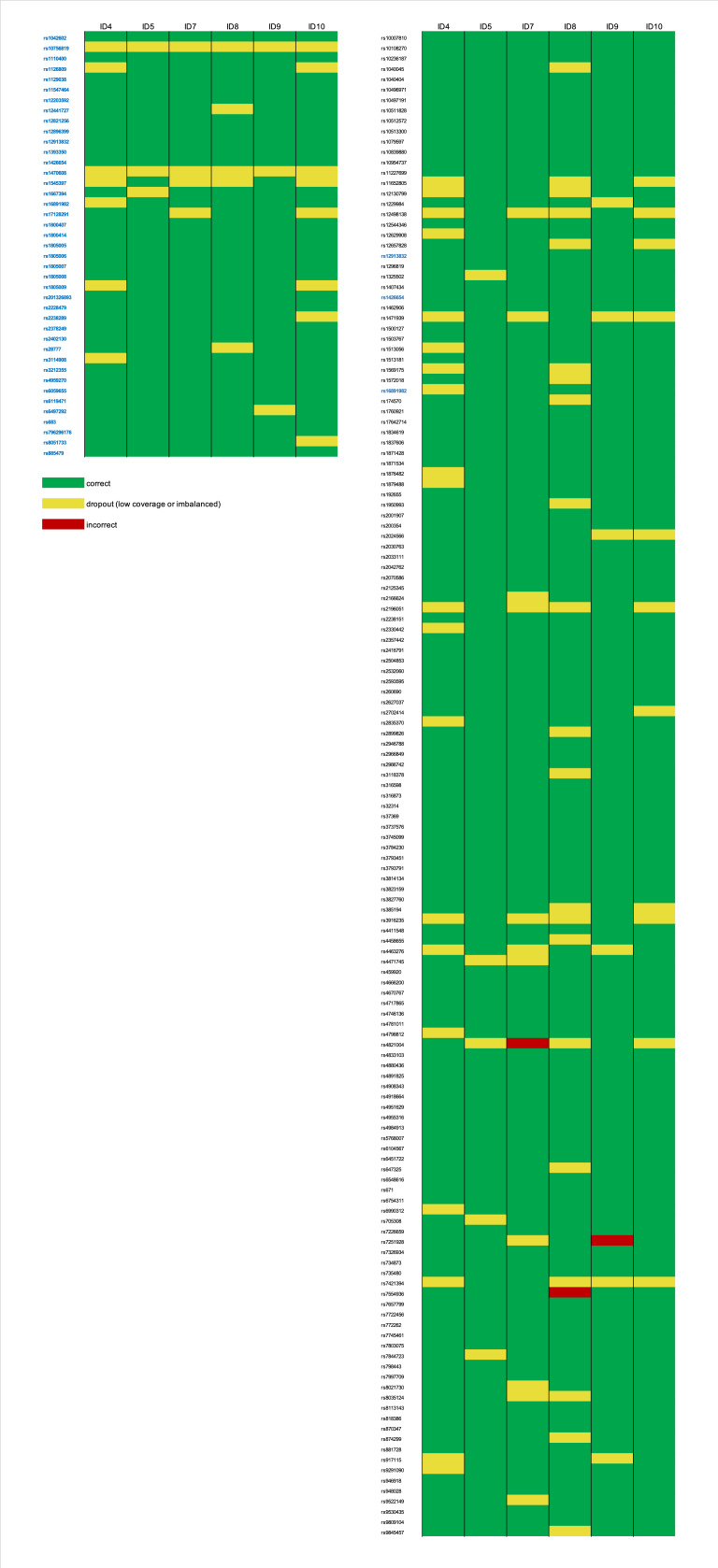
Genotyping summary for the selected single cells (with calling rate of at least 90%); left–41 SNPs used for the phenotype prediction, right–145 SNPs used for ancestry prediction.

**Table 1 genes-12-01362-t001:** Phenotype and ancestry prediction summary for cell groups and consensus profiles from single cells. *p*-values and admixture proportions were compared with the results obtained for reference (ref.) sample (*italic* indicates identical values).

	Ref.	Cells–Consensus
20	10	5	1 “basic”	1 “conservative”
Eye color	blue	**0.000**	*0.000*
inter	**0.002**	*0.002*
brown	**0.998**	*0.998*
Hair color and shade	blond	**0.003**	*0.003*
brown	**0.337**	*0.337*
red	**0.000**	*0.000*
black	**0.661**	*0.661*
light	**0.005**	*0.005*
dark	**0.995**	*0.995*
Skin color	very pale	**0.001**	*0.001*
pale	**0.008**	*0.008*
inter	**0.275**	*0.275*	0.284
dark	**0.696**	0.687	0.676
dark to black	**0.021**	0.029	0.030

Admixture	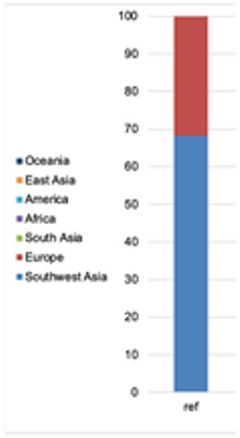	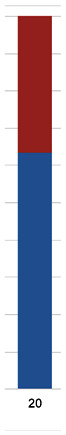	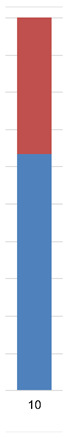	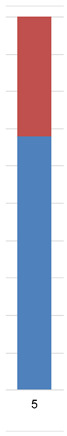	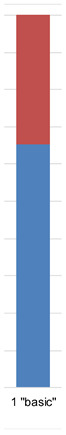	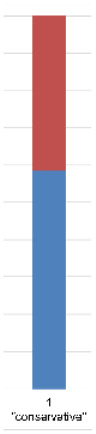

**Table 2 genes-12-01362-t002:** Phenotype and ancestry prediction summary for selected single cells (with calling rate of at least 90%). *p*-values and admixture proportions were compared with the results obtained from reference (ref.) sample (*italic* indicates identical values; * indicates different values that affected the interpretation of the results). AUC-Area Under Curve.

	Ref.	Single Cells	AUC Loss
ID4	ID5	ID7	ID8	ID9	ID10	ID4	ID5	ID7	ID8	ID9	ID10
Eye color	blue	**0.000**	*0.000*	0.003	0.000
inter	**0.002**	0.012	*0.002*	0.012	0.000
brown	**0.998**	0.988	*0.998*	0.003	0.000
Hair color and shade	blond	**0.003**	0.014	*0.003*	0.006	0.000	0.004
brown	**0.337**	0.521 *	*0.337*	0.378	*0.337*	0.343	0.007	0.000	0.003
red	**0.000**	*0.000*	0.028	0.000	0.027
black	**0.661**	0.465 *	*0.661*	0.619	*0.661*	0.654	0.002	0.000	0.001	0.000	0.001
light	**0.005**	0.018	*0.005*	0.003	*0.005*	0.001	0.000
dark	**0.995**	0.982	*0.995*	0.997	*0.995*	0.001	0.000
Skin color	very pale	**0.001**	0.003	0.000	0.000	*0.001*	0.012	0.004	0.002	0.003	0.002	0.002
pale	**0.008**	0.030	0.004	0.005	0.007	0.009	0.005	0.007	0.015	0.004	0.005	0.004	0.011
inter	**0.275**	0.418 *	0.153	0.172	0.252	0.278	0.187	0.003	0.012	0.001	0.002	0.001	0.008
dark	**0.696**	0.520 *	0.700	0.509 *	0.710	0.689	0.430 *	0.002	0.000	0.001	0.005	0.001	0.000
dark to black	**0.021**	*0.029*	0.143	0.315 *	0.030	0.024	0.377*	0.001	0.001	0.002	0.001	0.000	0.002

Admixture	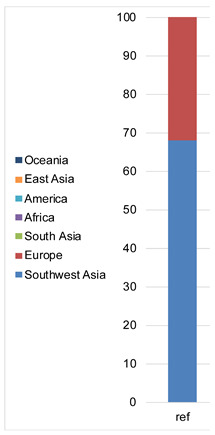	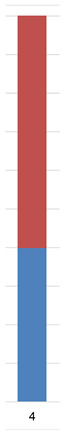	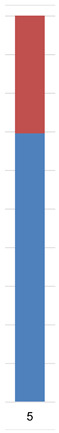	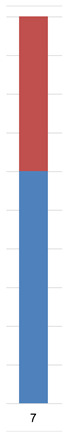	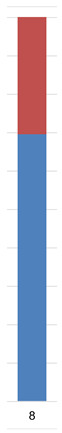	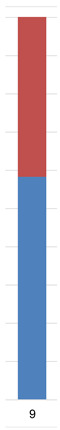	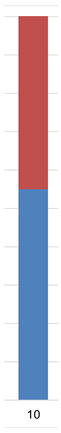	

## Data Availability

No data to report.

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
