# Peer review of "Pushing the Boundaries: Forensic DNA Phenotyping Challenged by Single-Cell Sequencing"

_genes, 2021, doi:10.3390/genes12091362_

Round 1

Reviewer 1 Report

Thank you for the opportunity to read and review the interesting and well-written manuscript called Pushing the boundaries: forensic phenotyping challenged by single-cell sequencing. This study is covering a new and interesting aspect in forensic genetics and the authors have set up a very well-planned study for this. The manuscript is covering single-cell sequencing and as something very novel, the downstream process of the sequencing is phenotypic traits instead of STR sequencing. The subject in the manuscript is relevant and well-explained. However, I have listed a number of minor suggestions below.

I know it might be difficult so long after the study was performed but it might make more sense to compare the obtained phenotypic results with what was observed from the donor of the DNA instead of comparing the results to other phenotypic results. This is not a major concern but it would make the setup slightly better.

When studying single-cell sequencing, a major concern will be contamination. I don’t see the authors address this here and with only one donor, it is not possible to uncover contamination issues with this study. The manuscript will benefit with some words about contamination if the authors are not concerned of this matter.

In line 273, “holopolymer” is written where I believe it is supposed to be “homopolymer”.

I figure 6, the x axis should be explained.

In Table 1, the “Ref.” should be explained. It is not clear to me what is meant here.

In figure 7 and table 2, the term “selected single cells” is used. The authors should explain more in detail what some cells were selected.

In table 2, the abbreviation “AUC” should be written out in the legend.

Numbers 1-10 should be written with letters.

Author Response

Thank you for the opportunity to read and review the interesting and well-written manuscript called Pushing the boundaries: forensic phenotyping challenged by single-cell sequencing. This study is covering a new and interesting aspect in forensic genetics and the authors have set up a very well-planned study for this. The manuscript is covering single-cell sequencing and as something very novel, the downstream process of the sequencing is phenotypic traits instead of STR sequencing. The subject in the manuscript is relevant and well-explained. However, I have listed a number of minor suggestions below.

We would like to thank the reviewer for a great feedback on our paper, we appreciate it a lot!

I know it might be difficult so long after the study was performed but it might make more sense to compare the obtained phenotypic results with what was observed from the donor of the DNA instead of comparing the results to other phenotypic results. This is not a major concern but it would make the setup slightly better.

We compare the obtained phenotypic results between the samples but overall, we do refer only to the results obtained from the donor, both in the manuscript and in the tables. In the latter, the p-values from the reference sample (called “ref.”) are in bold and we used colors to mark which p-values from the study were identical, similar or significantly different from the reference. 

When studying single-cell sequencing, a major concern will be contamination. I don’t see the authors address this here and with only one donor, it is not possible to uncover contamination issues with this study. The manuscript will benefit with some words about contamination if the authors are not concerned of this matter.

Thank you for the suggestion, we added explanation to the manuscript (Materials & Methods).

In line 273, “holopolymer” is written where I believe it is supposed to be “homopolymer”.

Typo corrected

I figure 6, the x axis should be explained.

Added

In Table 1, the “Ref.” should be explained. It is not clear to me what is meant here.

„Ref.“ means the reference sample (1ng input whole autopsy blood sample) from the same individual from which blood was used for the single cell study. Explanation is now added for both tables.

In figure 7 and table 2, the term “selected single cells” is used. The authors should explain more in detail what some cells were selected.

The term was explained earlier within the manuscript but detailed description is now added to both.

In table 2, the abbreviation “AUC” should be written out in the legend.

Explained

Numbers 1-10 should be written with letters.

Corrected

Reviewer 2 Report

The manuscript describes the use of single cell sequencing/group of cells to predict ancestry and HIrisPlex-S phenotypes (eye, hair, and skin colour). This is carried out by investigating blood in various quantities from a single donor using DEPArray for single cell isolation and the Ion Torrent for sequencing with the Pheno Trivium Panel. The topic is super exiting and the manuscript is overall well written and easy to follow, however I think it has been carried out a little bit in a rush, which is reflected by missing words, typos, quality of figures etc. This is though a minor thing.

The major advantage of using single cell sequencing in a forensic context is the potential use when material is scarce or when trace samples are found as mixtures of the same material (e.g. blood). This is also stated by the authors in the introduction (l.33-42). I would have liked very much if the study also included experiments with mixtures instead of only using blood from a single donor. The authors should have tried to mix blood samples from two individuals to create a mixture and then do the same experiment to see if they could separate the two donor and get the correct HIrisPlex-S and ancestry profiles.

The potential use of single cell sequencing for forensic DNA prediction is not fully answered after reading the manuscript. The study would greatly benefit form including such mixture studies.

Major comments:

The study uses fresh blood material. Fresh blood is not very likely to be found on a crime scene. The use of fresh material versus non-fresh decomposed material from crime scenes has to be discussed. The authors write L. 72. that only highly evaluated (of typical size, morphology 72 and staining) cells were chosen for the experiment. Is this expected to found in a trace sample.

Discuss the rationale for including cell groups (5x 20 cells, 10x 10 cells, 10x 5 cells)? In a real case with mixed blood samples from more than one individual, would such an approach be used? From my point of view, it is only the single cell that is of interest.

Three out of 10 single cells predicts skin colour correctly. As opposed to eye colour, skin colour is dependent on multiple markers with high effect to give the correct prediction. Eye colour is almost solely based on rs12913832. It should be discussed. Ancestry on the other hand is composed of many markers with similar low effect and is not greatly influenced is one or two markers is missing. This is also seen by the results in the study

You should be more specific when the term phenotype is used. It could refer to anything. In this case you most of the time use the term to describe the use of HIrisPlex-S prediction model to predict eye, skin, and hair colour.  

Fig.6 and Fig. 7 are very difficult to read due to the size

l.519: I only partly agree, since you carried out any experiments with mixtures. You have just showed that is it possible to get correct HIrisPlex-S prediction on material from a single cell. This should be rephrased

You frequently use the term, forensic phenotyping. It is more correct to call it forensic DNA phenotyping. This is also seen in the header of the article. I would correct this throughout the article.

Minor:

l.30: Next Generation Sequencing -> next generation sequencing (NGS). A reference here would also be appreciated.

l.33: It’s -> it is

l.40: …in forensic… -> …in the forensic…

l.44-45: The sentence is difficult to understand

l.46: forensic phenotyping -> forensic DNA phenotyping

l.118 & l.191: I would be more specific and use HIrisPlex-S instead of Phenotype in the subheading

l.172: Associated with the phenotype? Height, hair colour, blood pressure?

l.175: I assume, the authors mean heterozygote loci?

l.177: A homozygote loci cannot have minor alleles. Rephrase to the “The allele frequency for loci classified as homozygote was between 92-100…”

l.199: Which was the same as the reference -> Which was predicted to be the same as the reference

l.211: How are the calculations done? It is unclear for me if the reads were added together and the statistics were carried out on all samples. E.g. is the mean marker coverage, the mean marker per sample or is the mean marker the mean across all samples for a specific marker? Also, l.279-280

l.212: ca. -> approximately

l.212-220: Consider the number of digits. I would use the number without digits and round the numbers instead, because the reads are integers

Fig.1: In the text. It is the percentage of reads on target that is comparable and not the number of reads. You also write this in the next sentence, but please make this more clear.

Fig.3: You could consider sorting the loci according to the frequency

Fig.4: Se comment about homozygote loci (l.177). You could also consider to add information of n < 90% for each sample

l.302: You could carry out a statistical test (Fishers test) between the number of missing loci between NA and AMP

l.312: phenotype -> HIrisPlex-S prediction (correct this throughout the document)

Table 2: Selected single cells. How was this selection carried out? Random?...

l.415: More forensic DNA phenotyping panels exists, consider including Meyer et al. 2019 (https://doi.org/10.1016/j.fsigss.2019.10.058)

Author Response

The manuscript describes the use of single cell sequencing/group of cells to predict ancestry and HIrisPlex-S phenotypes (eye, hair, and skin colour). This is carried out by investigating blood in various quantities from a single donor using DEPArray for single cell isolation and the Ion Torrent for sequencing with the Pheno Trivium Panel. The topic is super exiting and the manuscript is overall well written and easy to follow, however I think it has been carried out a little bit in a rush, which is reflected by missing words, typos, quality of figures etc. This is though a minor thing.

We appreciate the feedback and we are happy that the reviewer finds our idea exciting! We tried to summarize lots of data coming from the study as a proof-of-concept paper and therefore it might look chaotic. We thank for the comments and hope that the revised manuscript reads better.

The major advantage of using single cell sequencing in a forensic context is the potential use when material is scarce or when trace samples are found as mixtures of the same material (e.g. blood). This is also stated by the authors in the introduction (l.33-42). I would have liked very much if the study also included experiments with mixtures instead of only using blood from a single donor. The authors should have tried to mix blood samples from two individuals to create a mixture and then do the same experiment to see if they could separate the two donor and get the correct HIrisPlex-S and ancestry profiles.

The potential use of single cell sequencing for forensic DNA prediction is not fully answered after reading the manuscript. The study would greatly benefit form including such mixture studies.

We absolutely agree that the real potential of this workflow can only be evaluated when tested on actual mixtures. The combined workflow using single cell selection with DEPArray and SNP sequencing with Ion S5 was not tested for forensics yet. Therefore, in order to evaluate the idea, we first present this study as a proof of concept. We not only present a novel workflow but also, we discuss in detail an additional step in the library preparation (library rescue) and we believe that our findings could be very useful for the forensic community, not only in case of single cells or forensic DNA phenotyping. Based on here presented results we have already started planning a follow up study in which we will analyze mock mixtures and degraded samples. We hope to be able to finish our experiments in the near future and share a more detailed insights on the use of single cell sequencing in deconvoluting mixtures submitted to FDP.

Major comments:

The study uses fresh blood material. Fresh blood is not very likely to be found on a crime scene. The use of fresh material versus non-fresh decomposed material from crime scenes has to be discussed. The authors write L. 72. that only highly evaluated (of typical size, morphology 72 and staining) cells were chosen for the experiment. Is this expected to found in a trace sample.

In this paper we refer to our publication (https://doi.org/10.1007/s00194-018-0291-1), which discusses single cell based STR typing and presents the results coming from real crime scene sample – 6 weeks old blood stain from a knife. The analysis was successful and resulted in full mixture deconvolution. Here presented results of SNP sequencing show even higher genotyping rate than in case of STRs. It sounds very promising however we agree that the workflow has to be challenged with degraded samples to prove its potential. Especially that, as we discuss in our earlier work, cell sorting works only if the sample in interest includes undamaged cells that can be colored with the antibodies. Therefore, as mentioned above, our upcoming experiments will include not only mixtures but also degraded samples.

Discuss the rationale for including cell groups (5x 20 cells, 10x 10 cells, 10x 5 cells)? In a real case with mixed blood samples from more than one individual, would such an approach be used? From my point of view, it is only the single cell that is of interest.

We agree that only the single cells would be analyzed if the workflow was to be implemented for mixed samples. We performed the experiments based on cell groups as the first stage of this study, in order to evaluate the performance of the combined DEPArray and Ion S5 workflow, as this was never done before. We presented the results here in comparison to the previous PhenoTrivium validation study and as discussed, the results were comparable and allowed us to continue with the single cells with more confidence. Additionally, most of the validation studies of various forensic SNP assays (including ours) use serial dilutions for the sensitivity and here we present the sequencing results from samples with exactly known DNA input as it was based on the number of cells collected.       

Three out of 10 single cells predicts skin colour correctly. As opposed to eye colour, skin colour is dependent on multiple markers with high effect to give the correct prediction. Eye colour is almost solely based on rs12913832. It should be discussed. Ancestry on the other hand is composed of many markers with similar low effect and is not greatly influenced is one or two markers is missing. This is also seen by the results in the study

Thank you for the suggestion, we discussed the prediction challenges in more detail.

You should be more specific when the term phenotype is used. It could refer to anything. In this case you most of the time use the term to describe the use of HIrisPlex-S prediction model to predict eye, skin, and hair colour.  

Agreed, corrected. We explained both in the abstract and within the manuscript, where the term „phenotype“ is mentioned for the first time (Materials & Methods), to what exactly we refer.

Fig.6 and Fig. 7 are very difficult to read due to the size

The figures are readable when opened in full size and this will be possible after the online publication. It is true for all the figures in the manuscript.

l.519: I only partly agree, since you carried out any experiments with mixtures. You have just showed that is it possible to get correct HIrisPlex-S prediction on material from a single cell. This should be rephrased

Agreed, rewritten

You frequently use the term, forensic phenotyping. It is more correct to call it forensic DNA phenotyping. This is also seen in the header of the article. I would correct this throughout the article.

Thank you for pointing this out, we realized this after submitting the paper and planned to correct it anyway.

Minor:

l.30: Next Generation Sequencing -> next generation sequencing (NGS). A reference here would also be appreciated.

Corrected and reference added.

l.33: It’s -> it is

Corrected

l.40: …in forensic… -> …in the forensic…

Corrrected

l.44-45: The sentence is difficult to understand

Agreed, the whole sentence is now rewritten.

l.46: forensic phenotyping -> forensic DNA phenotyping

Corrected within the whole manuscript.

l.118 & l.191: I would be more specific and use HIrisPlex-S instead of Phenotype in the subheading

Changed as explained above.

l.172: Associated with the phenotype? Height, hair colour, blood pressure?

Changed by explaining that the loci come from the HPS panel.

l.175: I assume, the authors mean heterozygote loci?

Yes, this is explained in the first sentence of the paragraph.

l.177: A homozygote loci cannot have minor alleles. Rephrase to the “The allele frequency for loci classified as homozygote was between 92-100…”

Rephrased

l.199: Which was the same as the reference -> Which was predicted to be the same as the reference

Rephrased

l.211: How are the calculations done? It is unclear for me if the reads were added together and the statistics were carried out on all samples. E.g. is the mean marker coverage, the mean marker per sample or is the mean marker the mean across all samples for a specific marker? Also, l.279-280

The total number of mapped reads and reads on target was calculated by running the CoverageAnalysis plugin on TorrentServer (for preliminary check) and the total coverage was calculated by adding together coverage for all the markers in Converge after running HIDGenotyper plugin (to call variants). Mean coverage across all 200 markers (from Converge) was calculated for each sample.

l.212: ca. -> approximately

Corrected

l.212-220: Consider the number of digits. I would use the number without digits and round the numbers instead, because the reads are integers

We corrected the numbers in the whole Results section.

Fig.1: In the text. It is the percentage of reads on target that is comparable and not the number of reads. You also write this in the next sentence, but please make this more clear.

Rephrased

Fig.3: You could consider sorting the loci according to the frequency

The chart shows loci sorted based on ascending avg. frequency.  

Fig.4: Se comment about homozygote loci (l.177). You could also consider to add information of n < 90% for each sample

We added minimum and maximum number of loci with <90% per sample.

l.302: You could carry out a statistical test (Fishers test) between the number of missing loci between NA and AMP

We calculated and added the statistic value.

l.312: phenotype -> HIrisPlex-S prediction (correct this throughout the document)

We kept the term “phenotype” because we explained at the beginning what do we refer to.

Table 2: Selected single cells. How was this selection carried out? Random?...

We explained it in 3.2.3. Data interpretation – single-cell based predictions, l. 488-489 (in the resubmitted manuscript).

l.415: More forensic DNA phenotyping panels exists, consider including Meyer et al. 2019 (https://doi.org/10.1016/j.fsigss.2019.10.058)

Thank you for this suggestion, we missed it! Added.

Round 2

Reviewer 2 Report

Dear authors,

You have replied nicely to my comments. The only thing I would suggest is to add proof of concent to your abstract.

Maybe change the beginning of l. 21 to: The results from this prof of concent study suggest....

Congrats with a very nice and interesting study